# CodeNet: A Large-Scale AI for Code Dataset for Learning a Diversity of Coding Tasks

**Ruchir Puri**[1], **David S. Kung**[1], **Geert Janssen**[1], **Wei Zhang**[1],
**Giacomo Domeniconi**[1], **Vladimir Zolotov**[1], **Julian Dolby**[1], **Jie Chen**[2,1],
**Mihir Choudhury**[1], **Lindsey Decker**[1], **Veronika Thost**[2,1], **Luca Buratti**[1],
**Saurabh Pujar**[1], **Shyam Ramji**[1], **Ulrich Finkler**[1], **Susan Malaika**[3], **Frederick Reiss**[1]

[1]IBM Research        [2]MIT-IBM Watson AI Lab        [3]IBM Worldwide Ecosystems

## Abstract

Over the last several decades, software has been woven into the fabric of every aspect of our society. As software development surges and code infrastructure of enterprise applications ages, it is now more critical than ever to increase software development productivity and modernize legacy applications. Advances in deep learning and machine learning algorithms have enabled breakthroughs in computer vision, speech recognition, natural language processing and beyond, motivating researchers to leverage AI techniques to improve software development efficiency. Thus, the fast-emerging research area of "AI for Code" has garnered new interest and gathered momentum. In this paper, we present a large-scale dataset *CodeNet*, consisting of over 14 million code samples and about 500 million lines of code in 55 different programming languages, which is aimed at teaching AI to code. In addition to its large scale, CodeNet has a rich set of high-quality annotations to benchmark and help accelerate research in AI techniques for a variety of critical coding tasks, including code similarity and classification, code translation between a large variety of programming languages, and code performance (runtime and memory) improvement techniques. Additionally, CodeNet provides sample input and output test sets for 98.5% of the code samples, which can be used as an oracle for determining code correctness and potentially guide reinforcement learning for code quality improvements. As a usability feature, we provide several pre-processing tools in CodeNet to transform source code into representations that can be readily used as inputs into machine learning models. Results of code classification and code similarity experiments using the CodeNet dataset are provided as a reference. We hope that the scale, diversity and rich, high-quality annotations of CodeNet will offer unprecedented research opportunities at the intersection of AI and Software Engineering.

## 1 Introduction

There is a growing trend towards leveraging AI for building tools that support software engineering and development [1, 2]. AI can manipulate and generate computer code, but can it do so with high quality? Many researchers are fascinated by this possibility, encouraged by AI successes in other domains and tantalized by the vision of computers programming computers. Some recent deep-learning models [3, 4] for code have received a lot of publicity: trained on vast amounts of data and using novel architectures with billions of parameters, they sometimes generate surprisingly plausible code.

Given the success of non-AI tools for code, why should we consider AI to augment or possibly replace them? Firstly, AI can help refine and re-tune the heuristics used by traditional coding tools. Secondly, based on the training data from past experience, AI can help prioritize when there is more than one sound answer [5]. Thirdly, an AI-based tool may handle incomplete or invalid code more

35th Conference on Neural Information Processing Systems (NeurIPS 2021) Track on Datasets and Benchmarks.

robustly, thus expanding its scope. Finally, AI can incorporate signals usually ignored by traditional tools for code, such as the natural language in identifiers or comments.

In the enterprise environment, developers often face code written by large teams over many years and geographies. Developers must manipulate such code to modernize it, fix bugs, improve its performance, evolve it when requirements change, make it more secure, and/or comply with regulations. These tasks are challenging, and it is crucial to provide tool support for developers to be more productive at performing them. It is well known that the latest advancements in deep learning algorithms rely on best-of-breed datasets, such as ImageNet, to create increasingly complex and powerful models. In this paper, we present "CodeNet", a first-of-its-kind dataset in scale, diversity, and quality, to accelerate the algorithmic advances in AI for Code.

To promote widespread adoption of CodeNet, we will be launching contests involving use cases based on the dataset. The first contest [6] will focus on diversity, inclusion and spurring interest among aspiring data scientists. We are partnering with the Global Women in Data Science organization (with presence in over 50 countries) founded by Stanford University [7] and targeting teams with at least fifty percent women. We are planning follow-up contests that target experienced AI practitioners.

The rest of the paper is organized as follows. Section 2 introduces the CodeNet dataset. Related datasets are discussed in Section 3, and the differentiation of CodeNet with respect to these related datasets is elaborated in Section 4. Section 5 describes how CodeNet was curated and Section 6 enumerates the usability features of CodeNet with several pre-processing tools to transform source codes into representations that can be readily used as inputs into machine learning models. Section 7 discusses the upcoming CodeNet contest and Section 8 describes important baseline experiments with the CodeNet dataset. Section 9 presents further uses of the CodeNet dataset and Section 10 concludes the paper.

## 2   The CodeNet Dataset

The CodeNet dataset consists of a large collection of code samples with extensive metadata. It also contains documented tools to transform code samples into intermediate representations and to access the dataset and make tailored selections. Our goal is to provide the community with a large, high-quality curated dataset that can be used to advance AI techniques for source code.

CodeNet is derived from the data available on two online judge websites: AIZU [8] and AtCoder [9]. Online judge websites pose programming problems in the form of courses and contests. The dataset consists of submissions to these problems, which are judged by an automated review process for correctness. Problem descriptions, submission outcomes, and associated metadata are available via various REST APIs.

**Scale and Statistics.** CodeNet contains a total of 13,916,868 submissions, divided into 4053 problems. Among the submissions, 53.6% (7,460,588) are accepted (compilable and pass the prescribed tests), 29.5% are marked with wrong answer, and the remaining rejected due to their failure to meet run time or memory requirements. To our knowledge, this is the largest dataset so far among similar kinds. Submissions are in 55 different languages; 95% of them are coded in C++, Python, Java, C, Ruby, and C#. C++ is the most common language, with 8,008,527 submissions (57% of the total), of which 4,353,049 are accepted. With the abundance of code samples, users can extract large benchmark datasets that are customized to their downstream use. See Figure 1 for a summary.

**Diversity.** The problems in CodeNet are mainly pedagogical and range from elementary exercises to sophisticated problems that require advanced algorithms. The submitters range from beginners to experienced coders. Some submissions are correct while others contain different types of errors, accordingly labeled. The submissions are in many different languages.

**Code Samples.** Each code sample is a single file and includes inputting the test cases and printing out the computed results. The file name uses standard extensions that denote the programming language, e.g., `.py` for Python. The majority of code samples contain only one function, although submissions to more complex problems might have several functions.

**Metadata.** The metadata enables data queries and selections among the large collection of problems, languages, and source files. The metadata is organized in a two level hierarchy. The first is the dataset level, which describes all problems. The second is the problem level, which details all the submissions to a single problem. Metadata and data are separated in the dataset structure.

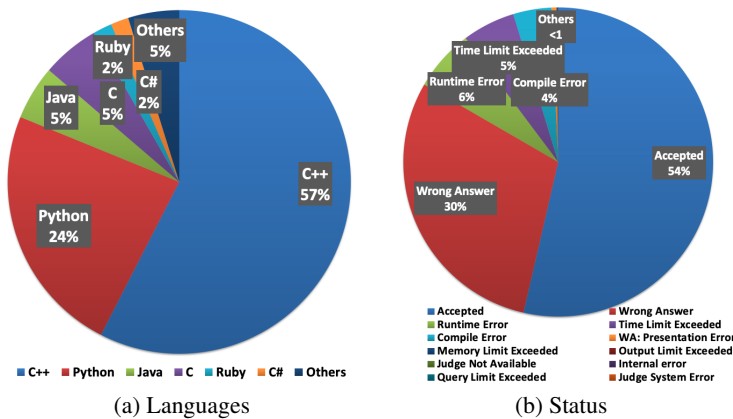

(a) Languages        (b) Status

Figure 1: Percentage of submissions per language (left) and per status (right).

At the dataset level, a single CSV file lists all problems and their origins, along with the CPU time and memory limits set for them. Additionally, every problem has an HTML file with a detailed description of the problem, the requirements and constraints, and the IO examples.

At the problem level, every problem has a CSV file. The metadata for each submission is summarized in Table 8 in the supplement, which lists the fields contained in each CSV file as well as the corresponding descriptions.

**Limitations.** All code samples in CodeNet may not be extensively commented, and these comments may be in multitude of languages. Therefore, AI techniques that rely on learning from preponderance of comments in the code may face challenges. The code samples are solutions to high-school and beginning college level programming problems. This dataset is not suitable for users looking for code with enterprise API's and advanced design patterns.

## 3 Related Datasets

A wide variety of datasets for source code exist, with many targeting one or a small number of tasks. Such tasks include clone detection, vulnerability detection [10, 11], cloze test [12], code completion [13, 14], code repair [15], code-to-code translation, natural language code search [16], text-to-code generation [17], and code summarization [16]. A detailed discussion of several of these tasks and their respective datasets is available in CodeXGLUE [18], which is a collection of existing datasets. CodeNet, on the other hand, is a new dataset curated from scratch, that aims to support a broad set of use cases. Popular datasets of a similar kind are POJ-104 [19] (which is incorporated as part of CodeXGLUE as well) and GCJ [20] (derived from Google Code Jam). We compare CodeNet to these datasets in the following.

### 3.1 POJ-104

POJ-104 was collected from a pedagogical online judge system. The code samples are submissions to 104 programming problems. With 500 submissions to each problem, there is a total of 52,000 code samples in the dataset. This dataset has been used by many authors for code classification [19] and code similarity [21].

POJ-104 is faced with several limitations.

1. The code samples are in C and C++, but the two languages are not distinguished. Although they are closely related, mixing them leads to parsing errors and a reduction of useful code samples [21].

2. Useful metadata such as the results of the judging system (acceptance, error types etc.) are missing. Therefore, for certain applications where compilabilty or code correctness is important, additional pre-processing efforts are needed and useful code samples are reduced [21]. The dataset does not contain the problem statement, although some example problems are described in [22], and information on how to execute the code samples is absent.

3. Some problems are identical (e.g., problems 26 and 62), and some submissions are near duplicates of each other, although the percentage of such cases is low compared to other datasets.

## 3.2 GCJ

GCJ [20] was collected from the submissions to the Google Code Jam competitions from 2008 to 2020. Similar to CodeNet, the submissions cover a wide variety of programming languages, with C++, Java, Python, and C being the predominant ones. The C++ subset has been extracted into a POJ-104-like benchmark and used in some publications. This benchmark dataset, GCJ-297 [23], has 297 problems and approximately 280K submissions. The number of submissions is imbalanced among problems.

GCJ is advantageous over POJ-104 in size and language diversity, but we believe that an even larger dataset such as CodeNet can better serve the community. GCJ contains neither metadata nor information on identical problems and near duplicates.

## 4  CodeNet Differentiation

Table 1: Related datasets comparison

|  | CodeNet | GCJ | POJ |
|---|---|---|---|
| Total number of problems | 4053 | 332 | 104 |
| Number of programming languages | 55 | 20 | 2 |
| Total number of code samples | 13,916,828 | 2,430,000 | 52,000 |
| C++/C subset data size (code samples) | 8,008,527 | 280,000 | 52,000 |
| Percentage of problems with test data | 51% | 0% | 0% |
| Task: Memory Consumption Prediction | Yes | No | No |
| Task: Runtime Performance Comparison | Yes | No | No |
| Task: Error Prediction | Yes | No | No |
| Task: Near duplicate prediction | Yes | No | No |

A high quality code dataset has certain desired properties. We constructed CodeNet according to these requirements. In the following, we discuss how CodeNet differentiates itself from the existing datasets along these lines. Table 1 is a comparison with related datasets.

**Large scale.** A useful dataset should contain a large number and variety of data samples to expose the realistic and complex landscape of data distributions one meets in practice. CodeNet is the largest dataset in its class - it has approximately 10 times more code samples than GCJ and its C++ benchmark is approximately 10 times larger than POJ-104.

**Rich annotation.** For the dataset class in question, it is important to include information beyond which problem a code sample solves to enable a wide range of applications and use cases. It is useful to know whether a code sample solves the problem correctly, and if not, the error category (e.g., compilation error, runtime error, and out-of-memory error). Since the source code is supposed to solve a programming problem, it is advantageous to know the problem statement and have a sample input for execution and a sample output for validation. All such extra information is part of CodeNet but absent in GCJ and POJ-104.

**Clean samples.** For effective machine learning, the data samples are expected to be independent and identically distributed (iid); otherwise, the resulting performance metric could be significantly inflated [24]. The existence of duplicate and/or near duplicate code samples makes the iid assumption dubious. Hence, it is crucial to identify the near duplicates. The presence of identical problems in the dataset poses an even bigger issue. In CodeNet, we analyzed the code samples for (near) duplication and used clustering to find identical problems. While this process does not make our dataset satisfy the iid property, providing this information as part of the dataset release allows more flexibility for the users to customize benchmarks for their specific use cases. The near-duplicate information is not available in GCJ and POJ-104.

## 5  Construction of CodeNet

### 5.1  Collection of Code Samples

The CodeNet dataset contains problems, submissions, and metadata, scraped from the AIZU and AtCoder online judging systems. For AIZU, we used the provided REST APIs to download all the

metadata. For AtCoder, due to the absence of a REST API, we scraped the problems, submissions, and metadata directly from the web pages. We considered only public and non-empty submissions that did not contain errors or inconsistencies in the metadata. We manually merged the information from the two sources and adopted a unified format to create a single dataset.

## 5.2 Cleansing

Because data are collected from different sources, we apply a consistent character encoding (UTF-8) on all raw data files. Additionally, we remove byte-order marks and use Unix-style line-feeds as the line ending.

As indicated in section 4, we identify near-duplicates. We follow Allamanis [24] and use Jaccard similarity [25] as a metric to score code pairs. Each code sample is tokenized and stored as a bag-of-tokens multiset. In our case, we keep all tokens except comments and preprocessor directives. We compute the set and multiset Jaccard indices and respectively use 0.9 and 0.8 as the near-duplicate thresholds.

Besides similar code samples, identical problems are also likely because they have been gathered over many decades. We go through the problem description files (in HTML format) and apply `fdupes` to extract identical problem pairs. Additionally, using the near-duplicate information calculated for code samples, we consider a problem pair to be a potential duplicate when the number of near-duplicate code pairs exceeds a threshold. Clustering of duplicate problems is illustrated by the graphs in Figure 2, where each node denotes a problem and an edge between two nodes is labeled by the number of near-duplicate code pairs. Each connected graph is then a cluster of potential duplicate problems and we manually inspect the problem descriptions to verify the correctness of this duplicate detection.

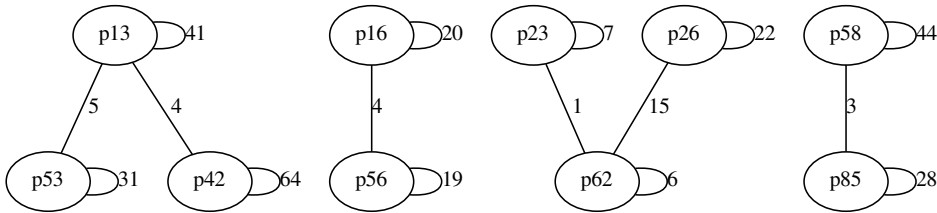

Figure 2: An example of a near-duplicate problem graph.

## 5.3 Benchmark Datasets

CodeNet has a rich set of code samples, and the user can assemble a customized benchmark according to his/her need. Following POJ-104, we extracted benchmark datasets from CodeNet in C++, Python, and Java. The benchmark characteristics are shown in Table 2. For the C++ benchmarks, the number of problems and their solutions are chosen to make the benchmark challenging. The benchmarks are filtered in the following ways. Each code sample is "unique" in the sense that it is not a near-duplicate of another code sample. The same is true of each problem. Samples with a large fraction of dead code are excluded. Each code sample has successfully passed through the tokenizer, the SPT generator, and the graph generator, all described in the next section. This step is to ensure that proper processing can be done to convert a code sample to a machine learning model input.

## 6 Code Representation and Tools

Machine learning with source code requires proper abstractions of the code. The abstractions are instantiated as representations in specific formats. As a usability feature, we provide several pre-processing tools to transform source codes into representations that can readily be used as inputs into machine learning models. They are described as follows.

**Tokenizer.** We offer fast C implementations of tokenizers for C, C++, Java, Python, and JavaScript. Additionally, the parse-tree generator described next can also produce token streams for C, C++, Java, and Python and can easily be extended to more languages.

**Simplified Parse Tree (SPT)** Simplified parse trees are derived from parse trees generated using ANTLR4 [26]. We traverse the ANTLR4 parse tree and remove internal nodes that only have one child. By doing so, we maintain the essential structure of the parse tree while pruning out unnecessary parser production rules. Finally, we adopt Aroma's [27] naming convention: leaf nodes are named by their literal strings and internal nodes are named by a concatenation of their children's names (only reserved words are kept while others are replaced by a hash mark #). We produce features for each node: (1) node type (token or parsing rule); (2) token type (e.g., an identifier), when applicable; (3) parsing rule type (e.g., an expression), when applicable; and (4) whether it is a reserved word. We adopt an extensible JSON graph schema so that edges can be augmented with types when needed. Currently, we support generating SPTs for four languages: C, C++, Java, and Python. Table 2 summarizes the SPT statistics for the four benchmarks.

Table 2: Benchmark statistics.

|            | C++1000     | C++1400     | Python800  | Java250    |
|------------|-------------|-------------|------------|------------|
| #problems  | 1,000       | 1,400       | 800        | 250        |
| #samples   | 500,000     | 420,000     | 240,000    | 75,000     |
| #SPT-nodes | 188,449,294 | 198,258,050 | 55,744,550 | 25,449,640 |
| #SPT-edges | 187,949,294 | 197,838,050 | 55,504,550 | 25,374,640 |

**Code graphs.** We augment the tool chain with a code graph generator using WALA [28], a general framework for program analysis. The backbone of a code graph is a system dependence graph, which is an inter-procedural graph of program instructions (e.g. call, read) expressing control flow and data flow information as edges. We also generate inter-procedural control flow graphs, which are control flow graphs of all the methods in the program, stitched together to connect call sites with target methods. Our code graph tool currently supports only Java and Python, but we plan to support more languages such as Javascript.

# 7    CodeNet Challenge

The launch of CodeNet was well received by the AI community and the media, with coverage from Forbes[29], VentureBeat[30], ZDNet[31] and others. Within a short span of 3 months, our github received 1000 stars and has been forked over 119 times. Our vision is to use CodeNet as an umbrella to curate AI for code datasets for widespread adoption and to drive innovation in AI for code. To leverage the momentum of CodeNet, we will be launching CodeNet challenges to create excitement in the AI community. The first contest [6] is mainly pedagogical and targets aspiring data scientists. In addition, we are partnering with the Global Women in Data Science organization (with presence in over 50 countries) founded by Stanford University [7] to emphasize diversity and inclusion (teams must have at least fifty percent women). We will organize workshops to introduce the topic, code similarity, and provide educational materials. This contest will be kicked off in late September and the winner will be announced in early December, around the NeurIPS2021 time frame. The conclusion of the first contest will be followed by a contest that will target experienced AI practitioners. Potential contest topics will revolve around practical and compelling use cases such as code language translation, code repair, code performance improvement, and code memory reduction.

# 8    Experiments with the CodeNet Dataset

In this section, we report the results of a code classification task, a similarity task, a generalization task, and a token inference task, using the four benchmark datasets (see Table 2) extracted from CodeNet. For this paper, these experiments are not meant to achieve the best-of-breed results using the state of the art. Our intention is to provide a set of baseline results as a reference. The experiments are typically performed on a Xeon machine using P100 or V100 GPUs. Details of the experiments are in appendices D, E, and F and their code and scripts are in the model-experiments folder of the CodeNet repository [32], when third party licenses allow.

## 8.1    Code Classification

In the classification task, each problem corresponds to a class: a code sample belongs to a class if it is a submission to the corresponding problem. For each experiment, 20% of the code samples are used for testing, while the rest are split in 4:1 for training and validation, respectively. We experiment with a diverse set of machine learning methods: bag of tokens, sequence of tokens, BERT model, and graph neural networks (GNNs).

1. **MLP with bag of tokens.** A code sample is represented by a vector of relative frequencies of token occurrences. Only operator and keyword tokens are used. The model is a 3-layer multilayer perceptron (MLP).

2. **CNN with token sequence.** We use the same set of tokens as above but retain their order to form a sequence. All sequences have the same length under zero padding. The classification model is a convolutional neural network (CNN) with an initial token embedding layer.

3. **C-BERT with token sequence.** Treating a code sample as a piece of natural language text, we build a C-BERT model [33] through pretraining on 10K top starred Github projects written in C. We use the Clang C tokenizer and Sentencepiece to tokenize each code sample. The pretrained model is fine-tuned on each benchmark.

4. **GNN with SPT.** Based on the parse tree representation, we use graph convolutional networks (GCN) [34] and graph isomorphism networks (GIN) [35] as well as their variants as the prediction model. The variant adds a virtual node to the graph to enhance graph message passing [36].

5. **GNN with Code Graph.** We also apply GCN on the code graph representation of the code.

Table 3: Classification accuracy (in %).

|  | Java250 | Python800 | C++1000 | C++1400 |
|---|---|---|---|---|
| MLP w/ bag of tokens | 71.00±0.29 | 67.80±0.15 | 68.26±0.21 | 64.50±0.13 |
| CNN w/ token sequence | 89.52±0.59 | 87.46±0.25 | 93.96±0.18 | 93.71±0.18 |
| C-BERT | 97.40±0.19 | 97.09±0.18 | 93.79±0.01 | 91.83±0.06 |
| GNN (GCN) | 92.70±0.25 | 93.82±0.16 | 95.76±0.12 | 95.26±0.13 |
| GNN (GCN-V) | 93.02±0.81 | 94.30±0.15 | 96.09±0.17 | 95.73±0.07 |
| GNN (GIN) | 93.26±0.23 | 94.17±0.19 | 96.34±0.15 | 95.95±0.13 |
| GNN (GIN-V) | 92.77±0.66 | 94.54±0.12 | 96.64±0.10 | 96.36±0.10 |
| Code Graph+GCN | 94.10±.001 | 87.80±.007 | N/A | N/A |

Table 3 summarizes the classification accuracy for all models on all benchmarks. Despite the simplicity of bag of tokens, it achieves well over 60% accuracy. Maintaining token ordering, CNN with token sequence offers significant improvement, reaching approximately 90% across all benchmarks.

More complex neural models sometimes further improve the prediction performance, as witnessed by C-BERT, which reaches approximately 97% for both Java and Python. It is interesting to note that even though C-BERT is pre-trained with C programs, its performance on the two C++ benchmarks is less impressive. We speculate that such a lower performance is related to programming practices. For C++, it is common to have identical program construction, such as declaration of constants (e.g., pi and epsilon) and data structures, appear across C++ submissions to different problems, but such a practice is rare in Java and Python.

Overall, the GNN models exhibit competitive performance. They are consistently the top performers, if not the best. The code graph representation slightly improves over the SPT representation on Java, but performs less well on Python.

## 8.2 Code Similarity

In the similarity task, two pieces of code samples are considered similar if they solve the same problem (type-4 similarity in [37]). Note that textual similarity does not guarantee similarity in functionality. For example, programs that differ by only one token might behave very differently; hence, they are not considered similar. For the token-based experiments, we treat the problem as binary classification. We use the same training, validation and testing split as in classification. Code pairs are randomly sampled within each subset. The number of similar pairs is the same as dissimilar ones. For the SPT representation, we experiment with several popular techniques, including AROMA [27], MISIM [21], and GMN [38]. The following contains more details about the models and methods.

1. **MLP with bag of tokens.** This model is the same as the one for code classification, except that the input is a concatenation of the two bag-of-tokens vectors from each program.

2. **Siamese network with token sequence.** The token sequence is the same as the one for code classification. The model is a Siamese network with two CNNs with shared weights.

3. **SPT with handcrafted feature extraction:** The method AROMA [27] uses normalized SPT node names and handcrafted rules to extract feature vectors for each SPT. Then, similarity is computed as a dot product of the extracted feature vectors.

4. **GNN with SPT:** With the same SPT, on the other hand, MISIM [21] uses a graph neural network to extract high-level features, and uses the cosine similarity of the extracted features to compute similarity. Additionally, we apply graph matching network (GMN) [38], which uses a cross-graph attention mechanism to learn pair-wise structural similarity of graphs, on the SPT pairs to predict similarity. The implementation is adapted from [39].

Table 4: Similarity accuracy (in %).

|  | Java250 | Python800 | C++1000 | C++1400 |
|---|---|---|---|---|
| MLP w/ bag of tokens | 81.80±0.06 | 86.61±0.08 | 85.82±0.05 | 86.54±0.07 |
| Siamese w/ token sequence | 89.70±0.18 | 94.67±0.12 | 96.19±0.08 | 96.56±0.07 |

Table 4 summarizes the classification accuracy for the first two models. The performance of bag of tokens is modest, considering that the problem is a binary classification with perfectly balanced classes. On the other hand, the Siamese model significantly outperforms bag of tokens, as expected.

Table 5: Similarity MAP@R score.

|  | Java250 | Python800 | C++1000 | C++1400 |
|---|---|---|---|---|
| Rule-based w/ SPT (AROMA) | 0.19 | 0.19 | 0.17 | 0.15 |
| GNN w/ SPT (MISIM) | 0.64±0.007 | 0.65±0.003 | 0.78±0.005 | 0.77±0.002 |

Table 5 summarizes the MAP@R [40] score for two SPT-based approaches with solutions for 50% problems used for training, 25% for validation, and 25% for test. MISIM GNN model is trained for 1000 epochs. AROMA results in a relatively low score because the feature extraction is rule-based and no model is learned, whereas MISIM uses a neural network to extract features through supervised training.

Table 6: Similarity MAP@R score on Java250.

|  | (p4, s5) | (p3, s300) | (p10, s300) |
|---|---|---|---|
| GNN w/ SPT (MISIM, structure only) | 0.472±0.023 | 0.194±0.010 | 0.096±0.009 |
| GNN w/ SPT (GMN, structure only) | 0.679±0.056 | 0.432±0.035 | 0.256±0.015 |
| GNN w/ SPT (GMN + MISIM node attributes) | 0.985±0.015 | 0.794±0.036 | 0.780±0.026 |

Exploring further into the Java250 benchmark, Table 6 summarizes the MAP@R score with a variety of test sets: (p4, s5), (p3, s300), and (p10, s300), indicating 4, 3, and 10 problems with 5, 300 and 300 solutions each respectively. Across all test sets, GMN outperforms MISIM if both are trained with only the SPT structure; when combined with MISIM node attributes, GMN further improves the score significantly.

## 8.3 Generalization Across Datasets

Models trained on the CodeNet benchmark datasets can benefit greatly from their high quality. To demonstrate this, we compare C++1000 to one of the largest publicly available datasets of its kind, GCJ-297 [23]. For the purpose of this comparison, we train the same MISIM model on C++1000 and GCJ-297 and test the two trained models on a third, independent dataset - POJ-104. The result of this comparison is plotted in Figure 3.

The $x$-axis of this plot is the number of training epochs used and the $y$-axis is the MAP@R score. The MISIM model for both datasets is trained for $500$ epochs and the MAP@R score for validation and test is computed after every ten epochs. There are a total of four curves - a validation and a test curve for GCJ-297 and a validation and a test curve for C++1000.

The training curves show that a $10\%$ higher validation score can be achieved with GCJ-297 compared to C++1000. However, when tested on POJ-104, the model trained on GCJ-297 achieves a $12\%$ lower score compared to the model trained on C++1000. We believe C++1000 has better generalization than GCJ-297 mainly for two reasons: i) high data bias in GCJ-297 because the top 20 problems with the most number of submissions account for $50\%$ of all submissions and ii) cleaning and de-duplication of submissions in CodeNet dataset (as described in Section 5.2).

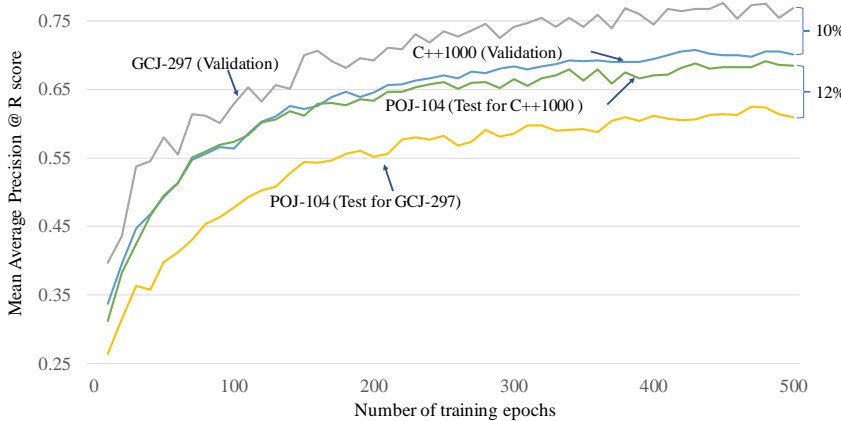

Figure 3: Test score on POJ-104 is 12% higher when a model is trained on C++1000 as compared to a model trained on GCJ-297, even though the validation score for GCJ-297 model is 10% higher than the validation score for C++1000 model.

### 8.4 Masked Language Modelling for Token Inference

A task such as code completion relies on the ability to predict a token at a certain position in a sequence. To accomplish this we can build a masked language model (MLM) using a technique that randomly masks out tokens in an input sequence and aims to correctly predict them in an as-yet-unseen test set. We train a popular BERT-like attention model on the C++1000 CodeNet benchmark after tokenization to a vocabulary of over 400 tokens and obtain a top-1 prediction accuracy of 0.9104 (stddev: 0.002) and a top-5 accuracy of 0.9935 (stddev: 0.0005).

## 9 Future Work and Experiments under Consideration

To explore and leverage the richness of CodeNet, we plan to consider the following experiments, some of which are in progress. Since the CodeNet dataset is multilingual, it is natural to perform code similarity across different programming languages. Our deep learning architecture for single language code similarity is a convolution Siamese neural network. To handle cross language similarity, the Siamese branches of the network do not share their weights. Moreover, the layers of those branches may have different sizes, as they process token sequences of different languages. Our first experiments achieved 73.5% accuracy for code similarity across C++ and Java, using the C++1400 and the Java250 benchmark datasets, and 87.49% accuracy across C++ and Python, using the C++1400 and the Python800 benchmark datasets. This cross-language similarity engine facilitates the construction of relationships between submissions within each language and across different languages, which will be useful in extracting supervised training datasets for automatic programming language translation.

Code repair [41, 42, 43] is the task of identifying and correcting programs that have errors, which is challenging, especially for logical errors. CodeNet provides a large number of correct code samples across many languages, which can serve as the basis for error injection to create large training datasets. Such synthesized error injections provide an excellent mechanism to create standardized validation datasets with controllable distributions of error types and difficulties. Furthermore, the rich metadata of CodeNet provide information about accepted and rejected submissions for the same problem by the same author in a chronological manner, which allows us to create 'good/bad' code pairs in a realistic setting. Using these datasets, we are investigating code repair using a transformer neural network.

We plan to expand our Graph Neural Network effort by applying GNN-based techniques to bridge the gap between natural language and source code for text-to-code applications such as code search, code completion and generation. To achieve this goal, we are exploring richer representations of both natural language and source codes along with a mix of supervised and unsupervised training strategies. We plan to exploit the pre-training/fine-tuning paradigm and leverage the foundational generative models like GPT-2 and GPT-Neo. Unsupervised pre-training can be performed on code samples from

the CodeNet dataset and data collected from community discussion forums like Stack Overflow. Our initial focus is on text to code generation and we plan to fine-tune GPT-Neo on a combination of the APPS dataset [44] and CodeNet, since CodeNet also contains problem specifications and test inputs.

## 10   Further Uses of CodeNet

The rich metadata and language diversity open CodeNet to a plethora of use cases. The problem-submission relationship in CodeNet corresponds to type-4 similarity [37] and can be used for code search and clone detection. A large number of code samples come with inputs so that we can execute the code to extract the CPU run time and memory footprint, which can be used for regression studies and prediction.

CodeNet may also be used for program translation, given its wealth of programs written in a multitude of languages. Translation between two programming languages is born out of a practical need to port legacy codebases to modern languages in order to increase accessibility and lower maintenance costs. With the help of neural networks, machine translation models developed for natural languages [45] were adapted to programming languages, producing pivotal success [4]. One considerable challenge of neural machine translation is that model training depends on large, parallel corpora that are expensive to curate [46], especially for low-resource languages (e.g., legacy code). Recently, monolingual approaches [47, 4] were developed to mitigate the reliance on parallel data, paving ways to build models for languages with little translation. Compared with current popular datasets (e.g., [4, 48]), CodeNet covers a much richer set of languages with ample training instances.

## 11   Conclusion

Artificial intelligence has made great strides in understanding human language. Computer scientists have been fascinated by the possibility and tantalized by the vision of computers (AI) programming computers. In this paper, we presented "CodeNet", a first-of-its-kind very large-scale, diverse and high-quality dataset to accelerate the algorithmic advances in AI for Code. This dataset is not only unique in its scale, but also in the diversity of coding tasks it can help benchmark: from code similarity and classification for advances in code recommendation algorithms, and code translation between a large variety of programming languages, to advances in code performance improvement techniques. We hope that the scale, diversity and rich, high-quality annotations of CodeNet will offer unprecedented research opportunities at the intersection of AI and Software Engineering.

## 12   Acknowledgements

We would like to acknowledge AIZU and AtCoder for making the code submissions publicly available. We would like to thank the IBM Data Asset eXchange team for providing a platform to host the CodeNet dataset. We would like to thank the Women in Data Science team at Stanford University and the IBM Call for Code team for their collaboration in launching the CodeNet challenge.

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
