# OpenReview forum: "CodeNet: A Large-Scale AI for Code Dataset for Learning a Diversity of Coding Tasks"
_NeurIPS.cc/2021/Track/Datasets_and_Benchmarks/Round2 — NeurIPS 2021 Datasets and Benchmarks Track (Round 2)_

### Official Review · Reviewer_9jmd · 2021-09-10
**CodeNet: A Large-Scale AI for Code Dataset for Learning a Diversity of Coding Tasks**

**Rating:** 8
**Confidence:** 5

**Strengths:**


Data set is large and diverse with respect to language and problems.
Having multiple solutions to the same problem allows comparing them, an important property usually lacking in SE datasets.

Data set contains structure test and runtime values, usually lacking in typical MSR (Mining Software Repositories) data sets, providing a unique value.

Source code usually needs some pre-processing to fit ML and some utilities are provided, easing use.

The authors are aware of the existence of duplicated problems and solutions and aim to cope with it.


**Weaknesses:**

The solutions are to a small self-contained small problem and do not resemble software development (e.g., as can be seen in open source projects).
While it makes them not representative and maybe less suitable to “AI for code”, the main stated goal, they complement work done in fields like MSR.
Please add a threat regarding the representability of the data set.

I browsed some of the solutions and in general they are written badly.
The high reported ratios of non compiling code and wrong answers support that.
Of course, this is not your responsibility but might pose a problem for use as samples for AI.
I assume that the solutions do not pass a code review.
If some of the solutions are reviewed (e.g., for winning a contest), it might be useful to provide this data as examples of a better code.

You note in the limitation the possible lack of documentation. Please add numbers to that. In 203 Python solutions to q2 in the mini data there are only 36 lines with the comment symbol ‘#’


**Additional Feedback:**

I’m not sure how and if to treat the cooperation with Global Women in Data Science, the challenge,  and the project’s stars from an academic point of view.
As a personal note, I am impressed.

The use of the data set for translation is tempting but it is not clear how to do it. After all, each problem has many solutions in the same language so which pairs should be matched?
A possible way is to use cycles
For example
 Python1 -> ToJava(Python1) -> ToPython(ToJava(Python1)) == Python1
Is that what you had in mind?



**Clarity:**

Paper is well written.


**Correctness:**

You, like any other in this field, cannot claim that your data set is a sample of IID.
It is not clear if such a distribution even exists, needless to say, what are its properties.
Please remove this claim.
Regardless, identifying near duplicates is important and well done for that!

Code similarity is not well defined.
I’m not sure that Jaccard similarity with the threhods you specified are a suitable metric.
Maybe you can provide the computed value (and some of the similar code) as the reader will be able to decide independently.

Reporting the accuracy on the code classification problem is rather meaningless.
It is a problem with a large number of classes and the distribution is probably imbalanced for most.
I suggest either reporting accuracy on one vs all for a suitable problem or use more suitable metrics like mutual information and entropy.

I think that the masking experiment suffers most from the code not representing “regular” code development.
I think the results will be very different on other data sets, without many solutions to the same problem and a very narrow domain and vocabulary.
I think you should add a warning here.


**Documentation:**

Is the test data a property of the question, not the solution?
Do you aggregate it and can use test data from one solution to solutions lacking it?
Where is the test data? I did not find it in the mini data set.

Where is the problem metadata (e.g., difficulty)? It is not in the problem list.
Can you add the problem description?
The title is not enough to understand the problem.


**Ethics:**

Did you receive a contest from the site's owners?
As for the problem creators and solution writers, how was that managed?
Clearly you could not ask for direct contents but was there a license with approval to make the data public?

I did not find the data sheet required by this track.


**Relation To Prior Work:**

Data set is compared to two other related data sets.

The main work in MSR (even the conference named so) is on open source projects. You should at least mention its existence otherwise readers from the ML community might thing the working on solutions data sets is the common path.


**Summary And Contributions:**

Code net provides a very large, diverse and classified set of solutions to similar coding problems.
Solutions are cleaned from similar cases, data set is compared to relevant data set.
Experiments are conducted on typical ML tasks for this data set.

---

> ### Author Response · Authors · 2021-09-28
> **Incorporating suggestions regarding code quality, mini dataset etc.**
>
> Thank you for your compliments and appreciating the unique values of CodeNet, as well as your feedback, which we will work to incorporate in the manuscript.
>
> We agree that code quality from a software engineering point of view is a complex topic, and there is a range of code quality represented in CodeNet. We feel providing the range of samples that are not compilable, fail some tests, exceed the runtime or memory provides important metadata for AI techniques to learn for improving code correctness, and its performance.  Thank you for the suggestion regarding winning code. We will work on incorporating the information regarding codes that passed the contest problem criterion into the metadata. Regarding code comments, we agree and will work on incorporating this information which can be readily derived.
>
> Thank you for appreciating the importance of identifying near-duplicates. As suggested, in the revised version, we will remove the claim regarding IID. Regarding the Jaccard similarity, we do not propose to use that metric for our similarity experiments; the Jaccard metric was merely used during curation of the derived benchmark datasets to weed out possibly duplicate submissions.
>
> Regarding classification experiments, we only report these results on the benchmark datasets, and these are fully balanced with respect to the number of samples per problem and so in that particular case, accuracy is a good metric. As suggested, although MSR represents a different thread of mining code, we will certainly acknowledge this and add: "Mining software repos (MSR) represents an unrelated but important area of research in software engineering and analytics." Regarding test data, indeed, the input/output test data pertains to each problem and is available for 98.5% of the problems in CodeNet. You are correct that sample test inputs/outputs are indeed missing from the mini dataset, but all metadata, including input/output and extensive HTML is available from the main dataset using the problem name. We will add this data to the mini dataset as well.
>
> Regarding license approval of the data, as you can appreciate, IBM takes the issue of ethical use of data very seriously and has established policy and controls for releasing any dataset. We had a thorough review with IBM IP counsel, who studied the legality of publishing both AIZU and AtCoder based material and confirmed that we have the rights to publish this data. Here is their confirmation verbatim: ``IBM represents and warrants it is the original author of the dataset and has the right to re-publish associated third-party code under open source license terms. IBM further represents and warrants it has the authority to grant the rights and licenses (CDLA Permissive v2.0) associated with the dataset to third parties.‘’ License CDLA v2.0 is spearheaded by the "The Linux Foundation" to enable collaboration on open data for AI and ML models which is closely aligned with NeurIPS Datasets and Benchmark track’s mission.
>
> Regarding the datasheet, we submitted it along with the supplementary information as per the submission guidelines, and hence it can be found in the attached zip file. The zip file contains a single PDF which perhaps confusingly starts off with a copy of the full paper text. Beyond that there is an appendix B (pp. 18 - 22) for the datasheet. Using cross-language similarity analysis to transform our dataset is a good approach for training ML models for supervised language translation. We are in the process of implementing such a relation between source code files written in two languages - a necessary requirement for training ML engines for code translation.
>
> Thank you again for your comments and appreciation!

---

### Official Review · Reviewer_wtwb · 2021-09-21
**CodeNet is a rich dataset with unimaginative benchmark tasks**

**Rating:** 5
**Confidence:** 5
**Correctness:** The dataset and small experiments see…

**Strengths:**

The dataset speaks for itself, is very rich and useful. It has been well-curated and de-duplicated and can likely enable excellent experiments.

**Weaknesses:**

The authors only study essentially one model task on this excellent dataset. They built a classification model. They could do many more interesting and novel things with even a simple classification model, for example, just a binary classifier given a problem and the code and tests determine whether a given submission passes. This is surely a task which can be interpreted as "programming language understanding". They also have a very large parallel corpus of solutions to a given problem, which could power a supervised code translation task. A related criticism is that the authors only scratch the surface of the literature and are missing many important citations and task definitions in the community which their data could support. I encourage the authors to browse https://ml4code.github.io/ and find many more past works to connect their dataset to in order to have a much richer discussion of their vision for the possibilities of this data.

**Additional Feedback:**

Thanks for the great dataset!

**Clarity:**

The paper is well-written though reads more like a blog post and could benefit from less verbose inspiration and more concrete connections to the work of the field.

**Documentation:**

The documentation appears sufficient and I was able to download and examine individual examples without trouble.

**Ethics:**

They mentioned licenses, which is the only ethical issue I can think of for this dataset.

**Relation To Prior Work:**

It clearly discusses past work related to other specific datasets but not to previous work on modeling. Please review https://ml4code.github.io/ and imagine more novel uses for the dataset!

**Summary And Contributions:**

CodeNet is a large dataset of code challenge websites for multiple programming languages, each problem of which has validating unit tests, and each submission of which has error message and performance metadata in addition to whether the sample code passed. This is a great resource for next generation modeling of source code, and can open avenues for bug fixing and even performance improvement suggestions. The authors presented essentially one benchmark classification task on this rich dataset.

---

> ### Author Response · Authors · 2021-09-27
> **Comments on experiments, AI tasks, references, and licensing.**
>
> We would like to thank you for appreciating CodeNet as a 'great dataset' for its quality and usefulness.  In the following, we would like to address your comments.
>
> In the paper, in addition to code classification, we have a detailed discussion on code similarity as well (Section 8.2). For this task, we discuss AI techniques like MLP, Siamese DNN,  the AROMA rule-based algorithm, MISIM's GNN with and without node attributes, and graph matching networks. In addition, we discuss two more tasks namely generalization of datasets (Section 8.3) and a masked language model for token inference (Section 8.4). The details of these experiments are captured in the appendix.  Through this rich dataset, our goal is to enable research in the open community on a broad array of AI for Code tasks, many of which we may not have been imagined yet.  Our experiments are also targeted toward checking the robustness and quality of the data. Currently, we are working on other AI for Code tasks, such as: cross-language similarity exploiting the multi-language nature of our dataset, including code translation;
> and automation of code debugging using transformer models, where we are leveraging the rich metadata of CodeNet to extract realistic pairs of buggy and fixed code. We will continue to release these models and results to the community via our Project_CodeNet github.
>
> As for scratching only the surface of the literature, we are keenly aware of the excellent resources in https://ml4code.github.io/ and have been exploring them since prior to starting Project CodeNet. The primary goal of our paper (as we outlined in the abstract) is to present a first of a kind AI for Code dataset with a baseline set of experiments, as opposed to a comprehensive survey of literature. We tried to cover related research efforts through the 54 references cited, and compared and differentiated CodeNet with respect to relevant dataset efforts as well.
>
> Regarding the comments around licensing of a dataset, CodeNet is licensed under very liberal terms of the CDLA v2.0 license which is specifically designed by "The Linux Foundation" for the purposes of open data for AI and ML. Leading open community and industry stakeholders (Linux foundation data & AI executive director, Microsoft Chief IP counsel, IBM Chief Scientist, Creative commons CEO, OpenUK CEO among other industry leaders) have endorsed this open data license https://www.linuxfoundation.org/press-release/enabling-easier-collaboration-on-open-data-for-ai-and-ml-with-cdla-permissive-20/.
>
> We hope our response addresses your comments. Thanks!

---

### Official Review · Reviewer_F3md · 2021-09-23
**A nice dataset of code**

**Rating:** 7
**Confidence:** 3
**Clarity:** The paper is well written with enough…

**Strengths:**

The dataset itself has the following strengths:

- Large scale. It has 13,916,828 code samples.
- Rich meta-information. It includes meta-information like execution time, memory usage.
- Multi-language. It covers 55 programming laugages.

With these strengths, it can be used for more tasks than other existing datasets.

**Weaknesses:**

I think a major weakness is that the code is only collected from online judges. This means the code is only used to solve some small-scale basic problems. Due to the same reason, the code does not include any API usage examples of some popular libraries either.
This makes the dataset less directly applicable to real-world software engineering problems.

**Additional Feedback:**

N/A

**Correctness:**

The major claims in the paper are correctly supported by the content of the dataset. The dataset is collected and cleaned in a sound way. The baselines in the benchmark are all reasonable baselines.

**Documentation:**

This dataset is well documented. The Github repo is well maintained. The license is clean.
Baseline models are provided in the code so I believe the benchmark is reproducible.


**Relation To Prior Work:**

The paper tells its difference with some similar work such as POJ-104 and GCJ. The major advantage of this dataset is being large-scale and multi-language and having rich metadata.

**Summary And Contributions:**

This paper introduces CodeNet, a large-scale code dataset for learning a diversity of coding tasks. The dataset consists of more than 14 million code samples in 55 different languages. It is the first-of-its-kind dataset in scale, diversity, and quality. With this dataset, the authors conduct diverse AI for code experiments such as code classification, code similarity prediction, and token inference.

---

> ### Author Response · Authors · 2021-09-28
> **Codenet code diversity**
>
> We would like to thank you for appreciating CodeNet's contributions and for your comments.
>
> Although source code files that constitute CodeNet are collected from online judging sites, it has a diversity of problems in terms of difficulty of tasks. Since real world enterprise tasks themselves are broken into smaller tasks of varying difficulty, the problems in CodeNet are a good proxy of granular tasks.  We quote one of the more complex contest problems here: 'In this programming contest, you will run a delivery service. Customers will place orders with your shop. Each order has a unique <var>\text{ID}</var> and should be delivered to the corresponding customer. Your delivery service has one car. The car will fetch the ordered item from the shop and deliver it to the customer.' And this contest has two variants, a static optimization problem in which all deliveries are known in advance, and a dynamic version where orders arrive as deliveries are happening.  Another such problem poses the challenge of simulating the behavior of an emacs-like editor to the participants. We believe that such problems provide a significant level of difficulty for AI for Code Algorithms, and at a granular level capture the difficulty of real world tasks to teach AI to code. Although as you correctly point out, CodeNet codes do not include API usage examples of some popular libraries, it captures varying level of problem difficulty to provide a significant opportunity and challenge to the AI for Code research efforts.  We further note that the standard libraries of the languages, which are used in the problems, themselves have significant functionality such as sophisticated containers (C++) and services like HTTP servers (Java, Python).
>
> Thank you again for your comments and appreciation!

---

### Official Review · Reviewer_ZNP6 · 2021-09-23
**CodeNet: A Large-Scale AI for Code Dataset for Learning a Diversity of Coding Tasks**

**Rating:** 7
**Confidence:** 3

**Strengths:**

As the authors have described very well in the paper, the CodeNet dataset includes a large collection of programs that makes it very relevant for research:
- Multiple programs, in multiple languages, that solve the same set of problems
- Lot of metadata on problem statement and submission outcomes
- Good spread of problem complexity and programmer expertise
- Good set of tools for researchers to get started with prototypes
- Rich set of example use cases under the model-experiments subdirectory


**Weaknesses:**

The key weaknesses of the dataset have already been called out by the authors under "Limitations" section on page 3:
1. The data is derived from submissions to two sites that host programming competitions and courses. What are the issues that arise from the types of problems that get posted here? For example, these are not typical programming tasks encountered in enterprise application development.
2. Competition problems are pretty constrained and self-contained. It would be helpful to see solution attributes like number of files (if applicable, since I'd assume submissions are probably limited single file), number of lines of code, code quality and structuring. Given the source of the data, architectural-level considerations, structuring of code into modules etc. or even interdependencies between source files reflecting factorings of the problem would all be missing from the samples.
3. Also, is there any quality check for the code that is included in the dataset? Quite a bit of the code that gets submitted is probably by novices. While that data is also useful for certain types of research, how can that data (expert versus novice submissions) be separated out?

**Additional Feedback:**

No additional feedback.

**Clarity:**

The paper is well written and the dataset creation steps, and description of its attributes and evaluations are easy to follow.

**Correctness:**

The dataset has been constructed in a sound way by combining submissions to two online coding websites, and narrowing down the sample to those programs that compile and pass test cases, and also meet runtime and memory requirements.

**Documentation:**

The Github site is well structured and easy to navigate. The salient information from the paper has been called out on the site for easy reference. Both, the dataset and the metadata, have been provided as zipped tarballs. The license is also available in the repo.

**Ethics:**

According to the terms of service of the AtCoder site (https://atcoder.jp/tos): "Ownership and copyright associated with the programs submitted by the Users through the Services shall belong to the Users themselves." Given that CodeNet has already been launched, I assume the authors already verified that it is ok to scrape the code submissions and include them in this dataset. Can the authors please check and reconfirm that this is ok? I will raise my rating subject to reconfirmation.

**Relation To Prior Work:**

This work's relation to prior work in the space, and the contributions have been clearly discussed.

**Summary And Contributions:**

I would like to thank the authors for their responses to my comments. I am updating my rating based on their comments about ownership and copyright concerns that I had.

Large dataset with coverage over multiple languages, with multiple solutions for the same problem statement.

Authors also list rich annotations, that can be used to customize the downstream ML explorations, for the code in the dataset:
1. Does the code sample solve the problem correctly? If not, is the error at compilation stage, at runtime or a memory error?
2. The source code is tied to the problem it is meant to solve, and this information is also available from the sites
3. Simple unit tests are also provided for 98.5% of the code samples -- this can be used to evaluate any code generation/learning tasks and for DRL applications.

Code preprocessing utilities to kick off explorations into ML models, and results of code classification and code similarity experiments using the dataset have been provided as reference.

---

> ### Author Response · Authors · 2021-09-26
> **Confirmation of dataset permissions**
>
> We would like to thank you for your constructive comments. We are glad that you found our dataset to be a strong contribution.
> Regarding the comments around licensing of dataset, we would like to confirm that IBM IP counsel did a thorough review of the  CodeNet dataset and approved the release of the open dataset. IBM has been a leader in spearheading ethical use of data, and has extensive controls to ensure that we do so. IBM IP counsel studied the legality of publishing both AIZU and AtCoder based material and confirmed that we have the rights to publish this data. Here is their confirmation verbatim: ``IBM represents and warrants it is the original author of the dataset and has the right to re-publish associated third-party code under open source license terms. IBM further represents and warrants it has the authority to grant the rights and licenses (CDLA Permissive v2.0) associated with the dataset to third parties.‘’ License CDLA v2.0 is spearheaded by the "The Linux Foundation" to enable collaboration on open data for AI and ML models which is closely aligned with NeurIPS Datasets and Benchmark track’s mission. Leading open community and industry stakeholders (Linux foundation data & AI executive director, Microsoft chief IP counsel, IBM Chief Scientist, Creative commons CEO, OpenUK CEO among other industry leaders) have endorsed this open data license https://www.linuxfoundation.org/press-release/enabling-easier-collaboration-on-open-data-for-ai-and-ml-with-cdla-permissive-20/.
>
> We really appreciate your willingness and consideration to raise the score.
>
> In the following, we capture our response to the other comments.
> We believe a major strength of CodeNet is the variety of problems in terms of difficulty of tasks and quality of its metadata. One of the fields of the metadata "rating" captures the degree of difficulty of the problem . Since real world enterprise tasks themselves are broken into smaller tasks of varying difficulty, the problems in CodeNet are a good proxy of granular tasks.   We quote one of the more complex contest problems here: 'In this programming contest, you will run a delivery service. Customers will place orders with your shop. Each order has a unique <var>\text{ID}</var> and should be delivered to the corresponding customer. Your delivery service has one car. The car will fetch the ordered item from the shop and deliver it to the customer.' And this contest has two variants, a static optimization problem in which all deliveries are known in advance, and a dynamic version where orders arrive as deliveries are happening.  Another such problem poses the challenge of simulating the behavior of an emacs-like editor to the participants. We believe that such problems provide a significant level of difficulty for AI for Code Algorithms, and at a granular level capture the difficulty of real world tasks to teach AI to code. In addition, since the solutions are judged on multiple quality dimensions---test sets, runtime, memory, etc---accepted solutions to these relatively difficult problems are high quality as well.  It is also easy to extract additional metadata beyond what is currently provided. We have extensive statistics about the dataset for internal use. Using aggregation and SQL scripts on our github (https://github.com/IBM/Project_CodeNet/tree/main/tools/aggregation-scripts and https://github.com/IBM/Project_CodeNet/blob/main/doc/HSQLDB.md), one can readily derive files per problem, average file size in bytes or lines with or without counting comments, and other user specified criteria. We will continue to enhance CodeNet metadata with the commonly sought out fields by the open community.
> Thank you for your consideration again!

---

### Decision · Program_Chairs · 2021-10-09

**Decision:**

Accept

**Comment:**

Most reviewers agree on acceptance. The comments by the only reviewer that tends to rejection were successfully addressed by the authors. I, therefore, recommend acceptance.